# Position: Single Paradigms Cannot Achieve General Artificial Intelligence — Integration of Symbolic, Connectionist, and Behaviorist Paradigms is Essential

## Abstract

This is a position paper arguing that a single artificial intelligence paradigm is insufficient to achieve general intelligence. The construction of general intelligence must integrate the three paradigms of symbolic reasoning, connectionist learning, and behaviorism. While significant progress has been made in fields such as deep learning, symbolic reasoning, and reinforcement learning algorithms, these methods each face inherent structural limitations in perception, reasoning, adaptation, and long-term stability. Starting from the functional requirements of general intelligence, this paper analyzes the complementary relationships between different paradigms in representation learning, structural abstraction, and long-term adaptation, emphasizing that general intelligence cannot be achieved by extending any single paradigm. Instead, it requires a coordinated integration of learning-based representation, rule-based abstraction, and behavioral mechanisms. Finally, this paper calls for viewing paradigm integration as a core scientific issue in the research of general artificial intelligence, rather than a temporary engineering solution.

## 1. Core Problems Addressed by Three Main AI Paradigms and Their Inherent Limitations

The research of artificial intelligence (AI) has long formed three mainstream paradigms represented by symbolism, connectionism, and behaviorism. Although these paradigms differ significantly in methodology and technical paths, they do not compete to solve intelligent problems at the same level. Instead, they attempt to address core challenges of different levels and functional dimensions in intelligent systems respectively. Understanding this is a prerequisite for discussing the implementation path of artificial general intelligence (AGI).

### 1.1. Symbolism

Symbolism's core contribution lies in the systematic modeling of explicit knowledge representation and logical reasoning capabilities. Through symbols, rules, logic, and procedural reasoning mechanisms, it can clearly express relationships between concepts, constraints, and causal structures, thereby supporting interpretable, verifiable, and auditable reasoning processes. This paradigm demonstrates irreplaceable advantages in mathematical theorem proving, knowledge reasoning, planning, and decision-making, essentially providing an abstract and normative cognitive ability for intelligent systems. However, symbolism also has fundamental limitations: symbolic systems cannot be automatically generated from the real world, and their knowledge construction is highly dependent on manual design; at the same time, symbolic rules are extremely sensitive to noise, uncertainty, and continuous changes. Once rules are incomplete or assumptions are violated, system performance often degrades rapidly. This "fragility" and "symbol grounding problem" make pure symbolic systems difficult to operate independently in open and dynamic real environments.

### 1.2. Connectionism

Connectionism approaches intelligent problems from a completely different direction. Represented by neural networks, connectionist methods excel at learning high-dimensional, continuous distributed representations from large-scale data, effectively handling perception, pattern recognition, and generation tasks. In recent years, the development of large models has further amplified connectionism's capabilities in language understanding, image generation, and cross-modal modeling, enabling machine systems to exhibit human-like "intuitive intelligence" at the surface behavioral level for the first time. However, this ability essentially stems from the fitting of statistical correlations rather than explicit structural understanding. Connectionist systems struggle to naturally express variable binding, logical combination, and causal

[1] Anonymous Institution, Anonymous City, Anonymous Region, Anonymous Country. Correspondence to: Anonymous Author <anon.email@domain.com>.

Preliminary work. Under review by the International Conference on Machine Learning (ICML). Do not distribute.

reasoning, and their generalization ability relies more on the similarity of data distributions than systematic generalization at the rule level. Additionally, the internal decision-making process of the model is highly implicit, lacking interpretability, making it difficult to meet the requirements for reliability and verifiability in high-risk scenarios.

### 1.3. Behaviorism

The behaviorist paradigm, especially methods represented by reinforcement learning (RL), focuses on the closed-loop learning problem of perception-action-feedback. Through continuous interaction with the environment, intelligent agents can gradually form effective strategies under unknown or incompletely modeled conditions, making behaviorism demonstrate unique value in control, games, and embodied intelligence. The essential advantage of RL is that it does not require a complete world model in advance but continuously approaches optimal behavior through trial and error. However, this paradigm also faces fundamental challenges: extremely low sample efficiency, making it difficult to bear the cost of large-scale trial and error in the real world; the design of reward functions is highly dependent on manual experience, and it is difficult to accurately characterize complex goals and value preferences; more importantly, RL without abstract and symbolic layer support has a search space that tends to grow exponentially with task complexity, thereby limiting its scalability and generalization ability.

## 2. Capability Requirements of AGI and the Inaccessibility of Single Paradigms

AGI is not the performance extreme in a specific task but a comprehensive capability system that can continuously adapt, transfer, and evolve in the open world. An AGI-oriented system must simultaneously possess the ability to perceive complex environments, abstract and express knowledge, reason about goals and constraints, and actively acquire new knowledge and revise its own cognition through actions. These capabilities are not optional modules but basic elements forming an intelligent closed loop.

### 2.1. Key Capabilities of AGI

**Perception and representation learning:** Extract meaningful concepts and relationships from multi-modal sensory data to build the foundation of the world model.

**Abstract reasoning and planning:** Conduct logical deduction, causal inference, and long-term planning, and handle counterfactual scenarios of "if...then...".

**Adaptive learning and continuous evolution:** Improve through experience in open environments to achieve lifelong learning without forgetting previous knowledge.

**Interpretability and trustworthiness:** The decision-making process is transparent to humans, able to explain reasons and accept review.

**Value alignment and safety:** Behaviors conform to human values and make ethically acceptable decisions under uncertainty.

**Efficiency and robustness:** Learn quickly with limited resources and maintain stability under distribution changes and adversarial attacks.

### 2.2. Symbolism: Dilemmas in Precise Reasoning and Knowledge Representation

Symbolism is based on formal logic, regarding intelligence as a transformation process of symbolic structures. Its core advantages include strong interpretability with transparent and traceable reasoning processes, combinatorial generalization ability to handle infinitely complex scenarios through symbol combination, and convenient knowledge injection to directly encode human experts' prior knowledge. Technologies such as expert systems, knowledge graphs, and automatic theorem proving all originate from this paradigm.

However, symbolism faces three structural bottlenecks:

**Symbol Grounding Problem:** The semantics of symbolic systems completely depend on external interpretations, lacking an inherent connection between symbols and the physical world. As Harnad pointed out, pure symbolic systems are like operators in the "Chinese Room", who can correctly process symbols but do not understand their true meanings. This makes it difficult for symbolic systems to handle perceptual data and independently learn concepts from primitive experiences.

**Knowledge Acquisition Bottleneck:** Building a complete rule base requires a lot of manual engineering, and domain experts need to explicitly encode tacit knowledge. Minsky once estimated that building a common-sense knowledge base requires millions of rules, which is almost infeasible in practice. More fundamentally, much human knowledge is difficult to express with precise rules. For example, "birds can fly" has exceptions such as penguins and ostriches, and traditional logic is difficult to handle such fuzziness and uncertainty.

**Fragility and Closure:** Symbolic systems perform excellently within their design assumptions but often fail completely when facing boundary cases. The early medical diagnosis expert system MYCIN could provide professional advice in specific fields but could not handle cases beyond

the knowledge base. This "fragility" stems from the discrete nature of symbolic systems—either fully matching the rules or being unable to reason at all.

### 2.3. Connectionism: Boundaries of Pattern Learning and Representation Capabilities

Connectionism simulates the distributed information processing of the brain through artificial neural networks, especially deep learning, which realizes hierarchical feature learning through multi-layer non-linear transformation. Its revolutionary achievements include surpassing human levels in ImageNet image recognition, achieving high-quality machine translation, and generating fluent natural language texts.

The core advantages of connectionism are: end-to-end learning to automatically discover features from data without manual design; fault tolerance and robustness, as distributed representation enables the system to tolerate noise and damage; generalization ability to generalize from training data to unseen samples. Large-scale pre-trained models have also demonstrated amazing few-shot learning and in-context learning capabilities.

But this paradigm also has deep limitations:

**Interpretability Crisis:** The decision-making process of deep neural networks presents a "black box" characteristic, making it difficult to trace specific reasoning paths. This constitutes a serious obstacle in high-risk fields such as medical diagnosis and autonomous driving—when the system gives an incorrect diagnosis, doctors cannot determine the source of the error; when an autonomous vehicle makes a dangerous decision, engineers are difficult to locate the cause of the failure. Although research on explainable AI (XAI) has made progress, there is still no fundamental breakthrough in the complete interpretation of deep networks.

**Data Dependence and Efficiency Issues:** Deep learning requires massive labeled data, and the training process consumes huge computing resources. Human children can learn new concepts with only a few examples, while neural networks need thousands or even millions of samples. This gap reveals the fundamental disadvantage of connectionism in sample efficiency. More importantly, neural networks learn statistical correlations rather than causal mechanisms, leading to unstable performance in out-of-distribution (OOD) scenarios.

**Lack of Compositionality and Systematicity:** Connectionist systems lack systematic compositional capabilities. Humans can understand "John loves Mary" and thus understand "Mary loves John", while neural networks need to learn these two patterns separately. Although modern large language models have shown certain compositional capabilities, they still frequently commit compositional fallacies in complex logical reasoning, indicating that they have not truly mastered the systematic rules of symbol combination.

### 2.4. Behaviorism: Challenges in Environmental Interaction and Adaptive Learning

Behaviorism realizes the dynamic interaction between intelligent agents and the environment through the RL framework, with the core idea of optimizing behavioral strategies through trial and error and reward feedback. DeepMind's DQN, AlphaGo series, and RLHF in OpenAI's GPT training all reflect the power of this paradigm.

The advantages of behaviorism are: autonomous exploration ability to discover complex strategies without preset rules; adaptability to dynamically adjust behaviors according to environmental feedback; handling sequential decisions and excelling in long-term planning and delayed reward optimization. These characteristics make it outstanding in fields such as games, robot control, and recommendation systems.

But behaviorism faces more severe challenges:

**Extremely Low Sample Efficiency:** RL agents usually need millions or even billions of environmental interactions to learn effective strategies. Although AlphaGo Zero could master Go through self-play, it consumed massive computing resources. In real physical environments, such trial-and-error costs are often unbearable—robots cannot drive randomly on real roads to learn traffic rules.

**Exploration-Exploitation Dilemma:** Agents need to balance between using known strategies and exploring new possibilities, but effective exploration is extremely difficult in high-dimensional state spaces. In sparse reward environments, agents may be unable to obtain effective feedback for a long time, leading to learning stagnation. Although methods such as intrinsic motivation and curiosity-driven have alleviated this to some extent, no fundamental breakthrough has been achieved.

**Safety and Verifiability:** RL strategies are difficult to formally verify and may exhibit unexpected behaviors. Although OpenAI's robot hand successfully solved the Rubik's cube, it was found to use "cheating" strategies that exploited vulnerabilities in the physics engine during simulation. This phenomenon of reward hacking may lead to catastrophic consequences in real applications. More importantly, RL systems lack explicit goal representation, making it difficult to achieve human-understandable value alignment.

From this perspective, no single paradigm can independently meet all the requirements of AGI. Relying solely on connectionism, the system can exhibit strong perception and

generation capabilities on large-scale data but lacks explicit reasoning and value constraints, making it difficult to maintain consistency and reliability in OOD scenarios; relying solely on symbolism, the system is highly rigorous in reasoning but cannot effectively cope with noise, uncertainty, and perceptual complexity in the real world; relying solely on behaviorism, the system can learn strategies through interaction but is fundamentally limited in learning efficiency and generalization ability without abstract and structural guidance. Therefore, the realization of AGI is not a matter of scaling up a certain paradigm but a problem of integrating capabilities across paradigms.

## 3. Complementarity of Paradigms: Bottlenecks of Single Paradigms Can Be Compensated by Other Paradigms

Furthermore, the three paradigms are not simply parallel but present highly complementary structural characteristics. The greatest advantage of symbolism—explicit structure and reasoning ability—is exactly what connectionism lacks most; while connectionism's ability in perception and representation learning makes up for symbolism's deficiencies in grounding and robustness. Similarly, behaviorism provides irreplaceable causal feedback and new knowledge sources for symbolic systems and neural models through exploratory actions in the real world.

This complementary relationship means that the integration of the three paradigms is not an artificial splicing but inherently required by the structure of intelligent problems themselves. RL without symbolic constraints often falls into invalid search, while the introduction of symbolic planning and high-level abstraction can significantly reduce the search space; symbolic systems without behavioral feedback are prone to solidify on wrong assumptions, and their rule systems can be continuously revised through interactive learning; neural models without structural guidance are prone to overfitting data distributions, while their generalization behavior can be more stable under the action of symbolic rules and causal constraints. This indicates that a truly scalable intelligent system must simultaneously mobilize the capabilities of multiple paradigms at different cognitive levels.

Specifically:

- Symbolism excels at reasoning but lacks perceptual grounding, requiring connectionism to provide a bridge from senses to concepts;

- Connectionism is proficient in learning but lacks reasoning structure, requiring symbolism to provide compositionality and interpretability;

- Behaviorism can adapt but is inefficient, requiring sym-

bolism to reduce the search space and connectionism to provide prior knowledge.

This complementarity is not a simple superposition of functions but a deep integration of mechanisms. Just as human cognition is not a simple series of perception, reasoning, and action, but a dynamic process where the three systems are closely coupled and interpenetrated.

The development history of neuro-symbolic AI confirms this judgment. Early attempts such as the CILP system (converting logic programs into neural networks) and KBANN (injecting rules into network initialization) demonstrated technical feasibility but were limited by the capabilities of neural networks at that time. In recent years, with the revival of deep learning, neuro-symbolic integration has become active again: DeepProbLog combines probabilistic logic with neural networks, Logic Tensor Networks realize differentiable optimization of logical constraints, and Neural Theorem Provers learn heuristic strategies for auxiliary proof.

These advances indicate that paradigm integration is not only theoretically necessary but also technically feasible. However, most existing works focus on algorithm improvements for specific tasks, lacking systematic exploration of general integration frameworks. This is the research direction advocated in this paper.

## 4. Unified Cognitive Framework for Synergistic Integration of Three Paradigms

The previous analysis suggests that no single paradigm can satisfy the full set of AGI requirements. Instead, their weaknesses are largely complementary: symbolic reasoning improves structure, reliability, and explainability; neural representation learning provides grounding and robustness; and interaction-driven learning supplies feedback, adaptation, and new data.

### 4.1. Complementary Roles

**Symbolism as a structural supervisor.** Symbolic rules and planning can (i) constrain neural and RL policies to avoid invalid search, (ii) provide explicit intermediate decisions, and (iii) enable post-hoc explanations by mapping model decisions to interpretable rules.

**Connectionism as the grounding engine.** Neural models turn high-dimensional sensory streams into compact representations, enabling (i) scalable perception, (ii) uncertainty-aware outputs that can soften brittle symbolic decisions, and (iii) transfer of learned representations across tasks and domains.

*Table 1.* Complementarity across capability dimensions and integration requirements.

| Capability Dimension | Symbolism | Connectionism | Behaviorism | Integration Requirement |
|---|---|---|---|---|
| Perception and Representation | ✕ Requires manual features | ✓ Automatic learning | △ Relies on state design | Neural perception + symbolic concepts |
| Abstract Reasoning | ✓ Precise deduction | ✕ Statistical correlation | △ Implicit strategy | Neuro-symbolic reasoning |
| Adaptive Learning | ✕ Static rules | ✓ Gradient optimization | ✓ Trial-and-error optimization | Continuous learning and knowledge update |
| Interpretability | ✓ Transparent and traceable | ✕ Black box | △ Difficult to interpret strategy | Symbolic explanation generation |
| Sample Efficiency | ✓ No data required | ✕ Massive data | ✕ Massive interactions | Knowledge-guided learning |
| Combinatorial Generalization | ✓ Systematic combination | ✕ Pattern matching | △ Strategy transfer | Neuro-symbolic combination |
| Value Alignment | △ Rule encoding | ✕ Implicit preferences | △ Reward design | Symbolic value constraints + neural optimization |
| Open Environment Adaptation | ✕ Closed assumptions | ✕ Distribution sensitivity | ✓ Dynamic interaction | Symbolic planning + neural perception + RL adaptation |

**Behaviorism as the adaptation loop.** Interaction and reward feedback provide (i) goal-driven learning signals, (ii) online correction when assumptions fail, and (iii) an embodied channel that continuously updates both neural representations and symbolic knowledge.

### 4.2. Unified Architecture

We propose a layered, tightly coupled framework aligned with a "perception–cognition–action" loop:

- **Perception & Representation Layer (connectionism):** encodes multi-modal inputs into distributed representations; receives top-down constraints (e.g., logic-regularization, rule-conditioned decoding) from reasoning.

- **Coordination Layer (hub):** performs bidirectional translation and control, including (i) representation-to-symbol extraction (rule induction, clustering, structured decoding) and (ii) symbol-to-control compilation (constraints, reward shaping, action masks).

- **Cognitive Reasoning Layer (symbolism):** maintains a symbolic memory (rules/graphs), performs planning and constraint-based reasoning, and outputs goals, sub-goals, and verifiable decisions.

- **Behavior Interaction Layer (behaviorism):** executes policies in the environment (e.g., hierarchical RL), collects trajectories and feedback, and reports signals for model and knowledge updates.

Information flows both ways: bottom-up perception activates concepts and candidate rules; top-down goals and constraints guide representation learning and exploration.

### 4.3. Core Challenges

- **Representation mismatch:** bridging discrete symbols and continuous vectors without losing semantics.

- **Learning coordination:** avoiding interference among rule injection, gradient learning, and trial-and-error updates.

- **Dynamic routing & resource allocation:** deciding when to rely on rules, when to learn from data, and when to explore.

- **Long-term stability:** preventing overfitting, brittle rule drift, and unsafe RL behaviors under distribution shift.

### 4.4. Technical Path: Large Models as the Integration Medium

Under current conditions, large models can serve as a practical intermediate layer because they unify multi-modal representations and natural-language symbols.

- **LLM-based representation↔symbol conversion:** generate structured text/rules from latent states, and embed symbolic constraints back into continuous spaces.

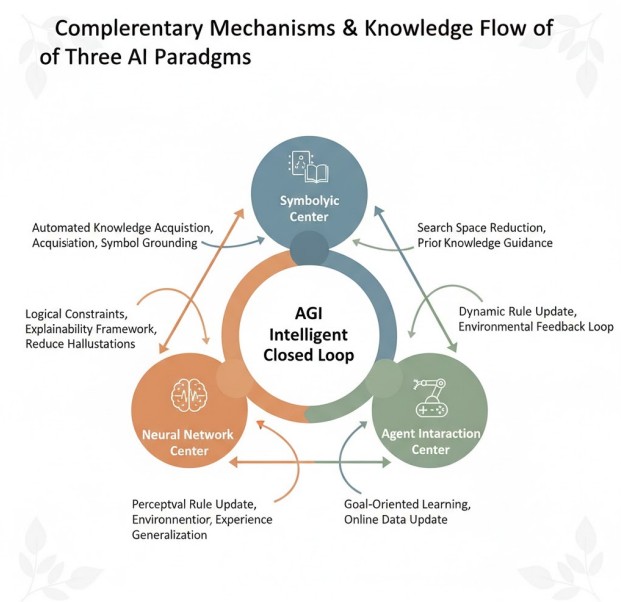

*Figure 1.* Complementary mechanisms and knowledge flow of three AI paradigms (symbolic, connectionist, and behaviorist) in an AGI closed loop.

- **Neuro-symbolic collaborative learning:** combine an LLM with an explicit reasoning module; use rule induction to distill symbolic constraints from model outputs and interaction traces.

- **Behavior–cognition closed loop:** use LLMs to decompose long-horizon goals into subgoals, compile them into constraints/rewards, and translate feedback into knowledge updates.

- **Generalization & robustness optimization:** use meta-learning/transfer and self-supervision to adapt across tasks while maintaining safety constraints.

## 5. Paradigm Reconstruction for AGI: Synergistic Evolution Beyond Traditional Forms

Under the proposed collaborative framework, each paradigm should be structurally reconstructed. Symbolism should evolve from a static, manually engineered rule base into a dynamic knowledge system that can be generated, learned, and revised; connectionism should focus on perception and representation while accepting explicit structural constraints; and behaviorism should move from blind search to efficient, goal-driven exploration guided by reasoning and world models.

### 5.1. Transformation of Symbolism: Lightweight, Generative, and Inferable Rule Framework

- **Lightweight rule representation:** use natural-language or structured rules that are easier to interface with large models; introduce fuzzy/probabilistic logic to handle uncertainty.

- **Automated rule generation:** extract rules from data and interaction via inductive logic programming (ILP), LLM prompting, and rule distillation.

- **Flexible reasoning:** adopt differentiable and incremental reasoning, combined with LLM context understanding for cross-scenario inference.

### 5.2. Transformation of Connectionism: Interpretable, Constrained, and Transferable Representation Models

- **Interpretable representation learning:** strengthen attention/feature attribution and build neuro-symbolic interfaces that map representations to rules.

- **Constrained training:** compile symbolic rules into regularizers/constraints; use behavioral feedback as a training signal to align representations with goals.

- **Transferable performance:** leverage pretraining, transfer, and meta-learning, combined with online updates for new environments.

### 5.3. Transformation of Behaviorism: Efficient, Guided, and Generalizable Interaction Mechanisms

- **Guided exploration:** reduce invalid trial-and-error using symbolic constraints, priors, and LLM-generated plans.

- **Adaptive rewards:** translate goals and values into reward shaping via rules and feedback, with LLMs assisting reward specification.

- **Generalizable strategies:** transfer abstract knowledge (symbols) and representations (neural features) to improve policy generalization.

## 6. Future Vision and Industry Call

### 6.1. Future Vision

Through synergistic integration and paradigm reconstruction, future AGI systems may achieve:

- **Trustworthy decision-making:** efficient perception with interpretable reasoning, explicit decision grounds, and safety under uncertainty.

- **Autonomous knowledge evolution:** acquire, verify, and refine knowledge through perception, interaction, and reasoning without heavy manual engineering.

- **Generalizable adaptability:** robust exploration and cross-task transfer in open, dynamic environments.

- **Human–machine symbiosis:** understand intent, collaborate effectively, and serve as an "intelligent partner" that augments human capabilities.

## 6.2. Industry Call

We call on the AI community and industry to:

- **Treat integration as a core AGI problem:** invest in representation conversion, learning coordination, and unified theory.

- **Promote cross-paradigm collaboration:** encourage interdisciplinary and academia–industry cooperation to accelerate deployment.

- **Emphasize paradigm transformation:** explore large models as integration hubs and break key bottlenecks toward interpretable, trustworthy, and generalizable systems.

## 7. Conclusion

This paper demonstrates that no single paradigm can achieve AGI, and that the only feasible path is the integration and collaboration of symbolism, connectionism, and behaviorism. We analyzed the strengths, inherent limitations, and bottlenecks of the three paradigms; summarized the core capability requirements of AGI; clarified their complementary mechanism; proposed a synergistic integration framework with large models as an intermediate medium; and discussed paradigm reconstruction directions, together with a future vision and industry calls.

AGI remains a long-term and challenging endeavor that will require sustained effort across generations. We believe that, by paradigm integration as the core direction, breaking key technical bottlenecks, and continuously improving the fusion framework and technical path, we can gradually approach general intelligence and enable transformative impacts on human society.

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
