# OpenReview forum: "Position: Single Paradigms Cannot Achieve General Artificial Intelligence — Integration of Symbolic, Connectionist, and Behaviorist Paradigms is Essential"
_ICML.cc/2026/Position_Paper_Track — Submitted to ICML 2026 Position Paper Track_

### Official Review · Reviewer_6wCE · 2026-03-08

**Significance:** 3
**Argument Clarity:** 2
**Rating:** 2
**Confidence:** 4

**Questions:**

1. In Section 1.1, the paper claims that "symbolic systems cannot be automatically generated from the real world". How do the authors reconcile this claim with the cognitive reality that human symbols and languages were, in fact, generated from the real world through evolutionary and developmental processes? Given that the symbolic emergence/origin of language has long been an active research field.

**Alternative Views Section:**

No

**Compliance With Llm Reviewing Policy A Conservative:**

Affirmed.

**Discussion Potential:**

1

**Final Justification:**

There is no rebuttal.

**Paper Summary:**

This position paper argues that achieving AGI requires an integrated paradigm of three foundational AI paradigms: symbolism, connectionism, and behaviorism. The authors reviewed the strengths and limitations of each paradigm across key capability dimensions necessary to achieve AGI, the paper concludes that no single paradigm is sufficient all the dimensions on its own. Finally, the authors propose an integrated framework that utilizes large models as an integration medium and call for paradigm integration to be treated as a core scientific issue rather than a temporary engineering solution.

**Position:**

Yes

**Position In Title:**

Yes

**Related Work:**

1

**Strengths And Weaknesses:**

### Strengths

1. The hybrid and integrated paradigm has long been a highly relevant and critical topic in the AI research.
2. The manuscript is well-structured, easy to follow, and it is good that the authors clearly articulates the complementary parts of the three paradigms into comparative dimensions (e.g., Table 1).

### Weaknesses

1. The central thesis of the hybrid paradigm and its importance to AGI have been widely discussed in the community for years. The discussions in the paper about the intersections remain largely superficial. For instance, the discussion on the relationship between neural and symbolic AI lacks technical depth and overlooks a significant body of recent, advanced literature in the neuro-symbolic domain [1].
2. For the hybrid paradigm itself, the paper does not introduce novel conceptual frameworks (e.g., a mathematical framework to describe different paradigms and there connections) or actionable insights (e.g., possible techniques and examples) that move beyond what is already actively being explored in the field. The proposed "Unified Architecture" using LLMs as an integration medium  is simply a summary of current trends rather than a distinct position, especially given that the paper does not provide any discussion on the alternative path (i.e., the non-LLM ways).
3. The paper heavily equates behaviorism with model-free RL, which misses the broader cognitive science aspects of behaviorism. Furthermore, since modern RL is deeply intertwined with deep neural networks (connectionism), the boundary drawn between behaviorism and connectionism in this paper feels somewhat artificial and reductive.

[1] Garcez, Artur d’Avila, and Luis C. Lamb. "Neurosymbolic ai: The 3 rd wave." Artificial Intelligence Review 56.11 (2023): 12387-12406.

**Support:**

2

---

### Official Review · Reviewer_bUQR · 2026-03-12

**Significance:** 2
**Argument Clarity:** 3
**Rating:** 3
**Confidence:** 4

**Questions:**

Are there other methods that could serve as the medium between different paradigms, aside from large language models?

**Alternative Views Section:**

Yes

**Compliance With Llm Reviewing Policy A Conservative:**

Affirmed.

**Discussion Potential:**

2

**Final Justification:**

No rebuttal

**Paper Summary:**

This paper investigates a core problem in general artificial intelligence, that a single paradigm is far from sufficient to support the realization of general artificial intelligence.

This paper points out that while single paradigms of artificial intelligence, such as symbolism, connectionism, and behaviorism, each possess their own strengths, they also exhibit inherent limitations. Regarding symbolism, constructing a comprehensive rule base necessitates substantial manual engineering, yet it often fails entirely when confronting boundary cases. Connectionism heavily relies on data, entails high training costs, and lacks interpretability. And behaviorism suffers from extremely low sample efficiency.

To address this challenge, this paper advocates for the integration of different, yet complementary artificial intelligence paradigms, wherein symbolism serves as a structural supervisor, connectionism acts as the grounding engine, and behaviorism functions as the adaptation loop.

**Position:**

Yes

**Position In Title:**

Yes

**Related Work:**

3

**Strengths And Weaknesses:**

Strengths:

This paper proposes a solution to a significant problem: integrating different yet complementary artificial intelligence paradigms to collectively support general artificial intelligence. The outcome is a layered, tightly coupled framework aligned with a "perception-cognition-action" loop, and it employs large language models as the medium for integrating these diverse paradigms.

Weaknesses:

1.The core challenges faced by frameworks integrating different artificial intelligence paradigms are insufficiently described.

2.The discussion on value alignment is insufficient.

3.In the Technical Path, only the scenario where large language models serve as a medium between different paradigms is discussed.

**Support:**

2

---

### Official Review · Reviewer_bfg1 · 2026-03-18

**Significance:** 1
**Argument Clarity:** 3
**Rating:** 2
**Confidence:** 4

**Questions:**

Why can't behaviourism alone solve all of the drawbacks that this paradigm has and achieve AGI? The authors' statements are based only on the current state of the research area, not on fundamental restrictions. Why is it principally not possible for a single paradigm to develop so much as to achieve "AGI"?

The authors proposed an AGI schema (Figure 1). This scheme looks like an abstract scheme of the any advanced Vibe-Coding system with tool calling, so nothing novel here as well. Why is it necessary to use three different concepts? Why can't I connect three Neural Network Centers (LLMs) that play the roles of the Symbolic Center and Agent Interaction Center to get the same results? If so, this means that Neural Networks are enough.

Why only these three paradigms?

The authors mentioned a list of problems that need to be solved for each of the areas in Section (4.3), and their conceptions are not helping to solve these problems.

What if we are building a human-in-the-loop AGI system? Maybe human contribution can close the gaps of each conception? Why is your suggested direction better than that?

**Alternative Views Section:**

No

**Compliance With Llm Reviewing Policy A Conservative:**

Affirmed.

**Discussion Potential:**

2

**Paper Summary:**

The paper describes the pros and cons of the different paradigms that can be applied to AI, showing that the positive parts of one intersect with the negative parts of another. So, by combining everything together, we can achieve a complex, General AI.

**Position:**

Yes

**Position In Title:**

Yes

**Related Work:**

1

**Strengths And Weaknesses:**

My main concerns are as follows. First of all, this position is not novel. It is one of the mainstream paradigms concerning the connection between Deep Learning and Symbolic systems. There is even a machine learning area named Neural-symbolic systems. Moreover, Causal Learning and Causal Reasoning can also be considered paradigms that combine several points of view. The only novelty of the authors is that they added RL there, but this is not a big deal. Moreover, we can consider behaviorism and its practical view (RL), as a part of connectionism. So, my general point is: this position is not novel. I believe, in the current state of AI, to actually move research into this Hybrid AI direction, there must be practical results, not another position, this position is well known.

There is another important note: Yes, in general I agree, in the current state, each paradigm has what they can do and what they don't. But, this is only the current state. Authors didn't show why fundamentally a single paradigm cannot develop so much as to achieve General Artificial Intelligence. So, the statement that "Single Paradigms Cannot Achieve General Artificial Intelligence" is not supported in any novel way.

Moreover, there is not citations in the paper, zero related work. A lot of points, items and paragraphs making it hard to read.

**Support:**

1

---

### Decision · Program_Chairs · 2026-04-30

**Decision:**

Reject

**Comment:**

The review process revealed significant concerns regarding the paper’s academic rigor and its contribution to the field, lacking sufficient novelty and technical insight. The proposition “hybrid systems are necessary” has been a mainstream paradigm, in particular well-established in the Neural-symbolic community. While the authors focus on the current technical limitations of individual paradigms, they did not provide deeper insight on the "fundamental restriction" or proof as to why a single paradigm could not principally evolve to solve these issues. In addition reviewers pointed out a critical lack of citations and "zero related work," which undermines the paper's standing as a formal position paper. Last but not least, the proposed AGI schema (Figure 1) was criticized for being too abstract and resembling existing "vibe-coding" or agentic systems with tool-calling, rather than offering a new architectural breakthrough.